# Arthritogenic Alphavirus Vaccines: Serogrouping Versus Cross-Protection in Mouse Models

**DOI:** 10.3390/vaccines8020209

**Published:** 2020-05-05

**Authors:** Wilson Nguyen, Eri Nakayama, Kexin Yan, Bing Tang, Thuy T. Le, Liang Liu, Tamara H. Cooper, John D. Hayball, Helen M. Faddy, David Warrilow, Richard J. N. Allcock, Jody Hobson-Peters, Roy A. Hall, Daniel J. Rawle, Viviana P. Lutzky, Paul Young, Nidia M. Oliveira, Gunter Hartel, Paul M. Howley, Natalie A. Prow, Andreas Suhrbier

**Affiliations:** 1Inflammation Biology Group, QIMR Berghofer Medical Research Institute, Brisbane 4029, Australia; wilson.nguyen@qimrberghofer.edu.au (W.N.); nakayama@nih.go.jp (E.N.); Kexin.Yan@qimrberghofer.edu.au (K.Y.); Bing.Tang@qimrberghofer.edu.au (B.T.); Thuy.Le@qimrberghofer.edu.au (T.T.L.); Daniel.Rawle@qimrberghofer.edu.au (D.J.R.); Viviana.Lutzky@qimrberghofer.edu.au (V.P.L.); 2Department of Virology I, National Institute of Infectious Diseases, Tokyo 162-0052, Japan; 3Experimental Therapeutics Laboratory, School of Pharmacy & Medical Sciences, University of South Australia Cancer Research Institute, SA 5000, Australia; Liang.Liu@unisa.edu.au (L.L.); Tamara.Cooper@unisa.edu.au (T.H.C.); john.hayball@unisa.edu.au (J.D.H.); 4Research and Development Laboratory, Australian Red Cross Lifeblood, Kelvin Grove, Qld 4059, Australia; hfaddy@usc.edu.au; 5Public Health Virology Laboratory, Queensland Health Forensic and Scientific Services, PO Box 594, Archerfield, Qld 4108, Australia; David.Warrilow@health.qld.gov.au; 6School of Biomedical Sciences, University of Western Australia, Crawley 6009, Australia; richard.allcock@uwa.edu.au; 7School of Chemistry and Molecular Biosciences, University of Queensland, St Lucia, Qld 4072, Australia; j.peters2@uq.edu.au (J.H.-P.); roy.hall@uq.edu.au (R.A.H.); p.young@uq.edu.au (P.Y.); 8Australian Infectious Disease Research Centre, Brisbane, Qld 4027 & 4072, Australia; 9Deptartment of Microbiology, University of Western Australia, Perth, WA 6009, Australia; n.oliveira@ucl.ac.uk; 10Statistics Unit, QIMR Berghofer Medical Research Institute, Brisbane, Qld 4029, Australia; Gunter.Hartel@qimrberghofer.edu.au; 11Sementis Ltd., Berwick, VIC 3806, Australia; p_howley@hotmail.com

**Keywords:** Chikungunya virus, Ross River virus, Mayaro virus, o’nyong nyong virus, Getah virus

## Abstract

Chikungunya virus (CHIKV), Ross River virus (RRV), o’nyong nyong virus (ONNV), Mayaro virus (MAYV) and Getah virus (GETV) represent arthritogenic alphaviruses belonging to the Semliki Forest virus antigenic complex. Antibodies raised against one of these viruses can cross-react with other serogroup members, suggesting that, for instance, a CHIKV vaccine (deemed commercially viable) might provide cross-protection against antigenically related alphaviruses. Herein we use human alphavirus isolates (including a new human RRV isolate) and wild-type mice to explore whether infection with one virus leads to cross-protection against viremia after challenge with other members of the antigenic complex. Persistently infected Rag1^-/-^ mice were also used to assess the cross-protective capacity of convalescent CHIKV serum. We also assessed the ability of a recombinant poxvirus-based CHIKV vaccine and a commercially available formalin-fixed, whole-virus GETV vaccine to induce cross-protective responses. Although cross-protection and/or cross-reactivity were clearly evident, they were not universal and were often suboptimal. Even for the more closely related viruses (e.g., CHIKV and ONNV, or RRV and GETV), vaccine-mediated neutralization and/or protection against the intended homologous target was significantly more effective than cross-neutralization and/or cross-protection against the heterologous virus. Effective vaccine-mediated cross-protection would thus likely require a higher dose and/or more vaccinations, which is likely to be unattractive to regulators and vaccine manufacturers.

## 1. Introduction

The arthritogenic alphaviruses that cause disease in humans include chikungunya virus (CHIKV), the Australian Ross River virus (RRV), the African o’nyong nyong virus (ONNV) and the South American Mayaro virus (MAYV) [1,2,3,4]. Symptomatic infection with these viruses results in a series of rheumatic manifestations that usually include acute fever, rash and myalgia and an acute, and often chronic, polyarthritis/polyarthralgia [4].

The largest outbreak of CHIKV-associated disease ever recorded began in 2004 and affected more than 100 countries (in Africa, Asia, Oceania, the Americas and Europe), with an estimated >10 million cases [5,6]. CHIKV disease also encompasses a spectrum of severe CHIKV disease manifestations resulting in hospitalizations (estimates range from 0.6% to 13% of cases) and mortality (estimates range from 0.024% to 0.7% of cases, primarily in the elderly) [5]. RRV causes a mean of ≈4600 cases per annum (2000–2015) in Australia where it is a notifiable disease [7], with a large outbreak (estimated at >60,000 cases) occurring in The Pacific Islands in 1979/80 [3,8]. In addition, recent outbreaks in the Shoalwater Bay Training Area in Australia (a major site of international military exercises) [9] highlighted the potential risk of international dissemination [10]. ONNV caused a large epidemic in 1959–1962 that started in northern Uganda and involved ≈2 million cases. A second epidemic was also reported in southern Uganda in 1996–7 [11]. However, subsequent case numbers have been low and sporadic [1,2], with malaria interventions against *Anopheles* species potentially also reducing ONNV transmission [12]. Nevertheless, ONNV morbidity in Africa is likely to be underestimated [12], and the risk of future outbreaks is considered high [13]. MAYV is largely restricted to South and Central America and the Caribbean, with about ≈30–100 cases per annum [1,2,4,13]. However, severe manifestations have been reported [14], and the emergence of recombinant MAYV strains represents a potential concern [15].

The large 2004–2019 CHIKV outbreak (and the potential severe disease manifestations) has resulted in the development of a series of vaccine candidates [5,16,17,18,19]. A CHIKV vaccine is deemed potentially [20] commercially viable [21], with the market size estimated at €500 million annually [5]. One CHIKV vaccine currently progressing into clinical trials is a recombinant poxvirus vaccine based on the multiplication-defective Sementis Copenhagen Vector (SCV) technology. The vaccine encodes the complete structural polyproteins of both CHIKV and Zika virus, and the vaccine is abbreviated as SCV-ZIKA/CHIK. The CHIKV polyprotein is used in many alphavirus vaccines [17,18,22,23,24] as self-assembled and matured viral surface glycoprotein spikes (comprising trimers of E1/E2 heterodimers) are believed to provide an authentically folded immunogen to the immune system [25,26,27]. One immunization with SCV-ZIKA/CHIK was previously shown to protect against viremia and disease after CHIKV challenge in an adult wild-type mouse model [16,28]. SCV-ZIKA/CHIK also induced neutralizing antibodies to CHIKV in non-human primates [29].

An RRV vaccine has been tested in a phase III trial and was well tolerated and immunogenic. This formalin and UV inactivated, whole virus RRV vaccine was alum-adjuvanted and delivered in three intramuscular 2.5 µg doses [22]. No further reports on development are publicly available, and the vaccine is currently owned by Ology Bioservices. Given the high cost of bringing a vaccine to the market [30] and the relatively low recognized case numbers for RRV, MAYV and ONNV, vaccines against these latter viruses are not likely to be deemed commercially viable.

The only commercial vaccine for an athritogenic alphavirus currently available is the formalin-inactivated, whole-virus Getah virus (GETV) vaccine that is sold by Nisseiken (Tokyo, Japan) as a mixed Japanese Encephalitis (JEV) and GETV vaccine [31]. This JEV/GETV vaccine is used in Japan to protect racehorses from GETV disease [31,32,33], which usually involves a 1–2 week-long, self-limiting disease characterized by fever, hind limb edema, lymph node swelling and a rash [34]. GETV has a broad geographical distribution that includes Asia, Europe and Australia [35] and was recently isolated from cattle in China [36]. RRV is also well known to infect horses [37,38], with some evidence for musculo-skeletal disease [39] and long-term poor performance [40].

Traditionally alphaviruses have been classified into antigenic complexes based on antibody cross-reactivity by hemagglutination inhibition, complement fixation and/or neutralization tests, with CHIKV, RRV, MAYV, ONNV and GETV all belonging to the Semliki Forest virus antigenic complex [41]. Consistent with this serogrouping, antibodies induced by infection with one of the aforementioned alphaviruses can often cross-react with other member(s) of this antigenic complex. For instance, (i) mouse convalescent RRV serum provided partial protection against CHIKV infection in wild-type mice [42], (ii) human convalescent CHIKV serum was able to cross-neutralize MAYV in vitro and in vivo, [43,44], (iii) CHIKV neutralizing monoclonal antibodies protected against ONNV in type I interferon-receptor-deficient mice and MAYV in wild-type mice [45] and (vi) MAYV-neutralizing monoclonal antibodies neutralized RRV and CHIKV in vitro [46]. An ensuing contention suggests that a vaccine for one of these alphaviruses might provide cross-protection against other viruses in the same antigenic complex [1,12,43,45]. In support, a live attenuated CHIKV vaccine was able to cross-protect against ONNV challenge in A129 mice [47], and the aforementioned RRV vaccine provided partial cross-protection against CHIKV in wild-type mice [48]. This contention perhaps finds support in the observations that cross-protection can be observed even for alphaviruses from different antigenic complexes [49,50,51]. However, the contention is not supported by the inconsistent cross-protection reported between relatively much more closely related CHIKV genotypes [52].

Using a newly identified human isolate of RRV from a transfusion case, RRV_TT_ [53] characterized herein, and human isolates of CHIKV, ONNV and MAYV, we investigate the ability of infection with one alphavirus to mediate cross-protection against viremia in adult wild-type mice after subsequent infection with a heterologous alphavirus. Antibody-dependent cross-protection was also assessed by adoptive transfer of CHIKV convalescent sera into Rag1^-/-^ mice persistently infected with the other three alphaviruses. Two contrasting vaccine modalities were also evaluated for their ability to mediate cross-protection; the live, multiplication defective, recombinant poxvirus SCV-ZIKA/CHIK [16,54] and the formalin-inactivated, whole-virus JEV/GETV vaccine [31,32,33]. Overall, our results argue that cross-protection, although clearly evident, is often suboptimal and unlikely to be of sufficient magnitude to be a realistic consideration in alphavirus vaccine development.

## 2. Materials and Methods

### 2.1. Ethics Statement

All mouse work was conducted in accordance with the “Australian code for the care and use of animals for scientific purposes” as defined by the National Health and Medical Research Council of Australia. Mouse work was approved by the QIMR Berghofer Medical Research Institute animal ethics committee (P2235 A1606-618M), with CHIKV work conducted in a biosafety level-3 (PC3) facility at the QIMR Berghofer MRI (Australian Department of Agriculture, Water and the Environment certification Q2326 and Office of the Gene Technology Regulator certification 3445).

### 2.2. Mice, Infection, Virus Titration and Disease Evaluation

Female C57BL/6J mice (6–8 week) were purchased from Animal Resources Centre (Canning Vale, WA, Australia). All other mice strains were bred in-house at QIMR B. Interferon Response Factor 3 and 7 deficient (IRF3/7^-/-^) mice [55] were kindly provided by M.S. Diamond (Washington University School of Medicine, St. Louis, MO, USA). Rag1^-/-^ mice are B and T cell-deficient on a C57BL/6 background; B6.129S7-*^Rag1tm1Mom^*/J (JAX). Rag2^-/-^ mice are B, T and NK cell-deficient on a B6 background; B6-*Rag2^tm1Fwa^ Il2rg^tm1Wjl^* (Taconic, Hudson, NY).

Mice were infected with 10^4^ CCID_50_ of the indicated alphavirus subcutaneously (s.c.) into the top/side of each hind foot as described previously [42,56]. The procedure is illustrated in Appendix A. Serum viremia, tissue titers and foot swelling were determined as described previously [42,55,56,57]. Mice were monitored using a score card system that provided a score for joint swelling, posture perturbations, loss of activity, fur ruffling/shivering (indicative of pyrexia), hind leg weakness, injection site reaction (1 = mild, 2 = moderate, 3 = severe); mice reaching a score of 2 for any two disease manifestations or 3 in any 1 disease manifestation were humanely euthanized using carbon dioxide.

### 2.3. Cell Culture

The African green monkey kidney, Vero (ATCC#: CCL-81), and the baby hamster kidney, BHK-21 (ATCC# CCL-10), cell lines were maintained in RPMI 1640 (Thermo Fisher Scientific, Scoresby, VIC, Australia) supplemented with endotoxin-free [58] 10% heat-inactivated fetal bovine serum (FBS; Sigma-Aldrich, Castle Hill, NSW, Australia) at 37 °C and 5% CO_2_. The *Aedes albopictus* mosquito-larva-derived cell line, C6/36 (ATCC# CRL-1660) was cultured in RPMI 1640 with 10% heat-inactivated FBS at 28 °C and 5% CO_2_.

### 2.4. The Arthritogenic Alphaviruses

RRV: RRV_T48_ has been described previously [59,60] (GenBank: GQ433359). RRV_TT_ was isolated (using C6/36 cells) from a human blood donation sample implicated in a case of transfusion transmission [53] (GenBank: KY302801.1).

CHIKV: the CHIKV (isolate LR2006-OPY1; GenBank KT449801.1; DQ443544.2) was isolated from a traveller returning from Reunion Island [61] and was a kind gift from Dr. P. Roques (CEA, Fontenay-aux-Roses, France).

ONNV: The ONNV (isolate IMTSSA/2004/5163, GenBank: DQ383272, DQ383273 and DQ399055) was isolated from a 19-year-old French soldier in Chad [62]. An infectious clone was constructed by Dr. A. Merits (University of Tartu, Tartu, Estonia) [63]; ONNV virus was derived from this clone and was provided by Dr G. Pijlman (University of Wageningen, Wageningen, Holland). The virus was imported under Australian DAWR permit no. 0002142850 and was classified as a BSL2 (PC2) organism.

MAYV: The MAYV (isolate BeH407, GenBank: QDL88200.1) was isolated from a febrile patient from the Guama River outbreak in Brazil in 1955 [64]. The virus was a kind gift from Dr. M. Diamond (Washington University School of Medicine, St. Louis, Missouri, USA) and was imported as for ONNV.

GETV: GETV_MM2021_ was isolated in Malaysia from *Culex gelidus* in 1955 and was kindly provided by Dr. J. Aaskov of the WHO Collaborating Centre for Arbovirus Reference and Research, Queensland University of Technology, Brisbane, Australia and was re-sequenced to provide a full genome sequence (GenBank MN849355) and to confirm the identity of this isolate that was provided to the Reference Centre in the 1960s.

All the alphaviruses [65] and cell lines (MycoAlert, Lonza) tested negative for mycoplasma. All the alphaviruses, except ONNV, were propagated in C6/36 cells and were titered by CCID_50_ assays as described using C6/36 and Vero cells [42]. ONNV was propagated in BHK-21 cells and titered by CCID_50_ assays using BHK-21 cells. All medium and fetal calf serum were negative for endotoxin contamination using a highly sensitive macrophage reporter cell line [58]. In our experience, alphaviral preparations with even low levels of endotoxin contamination replicate poorly in wild- type mice. Viral expansion in the indicated cells is shown in Appendix A.

### 2.5. RRV_TT_ Sequencing and Phylogenetic Tree Construction

RRV_TT_ genome sequencing was performed as described previously [66]. Briefly, sequence-independent amplification was performed on viral RNA extracted from the isolate grown in C6/36 cells. A library was prepared from the amplified products and sequenced using a 316 chip on the Ion Torrent PGM platform, generating >500,000 reads. A consensus sequence was generated by performing an assembly with Geneious R8 software (Geneious, Auckland, NZ https://www.geneious.com) using RRV T48 (GenBank: GQ433359) as a reference. An alignment was performed with genome-length (and near-genome-length) RRV virus GenBank submissions using the MAFFT plugin of Geneious R8 with default parameters. A phylogenetic tree was constructed with the nucleotide sequence alignment in MEGA7 (Molecular Evolutionary Genetics Analysis version 7.0, Penn State University, Pennsylvania, USA) [67] using the Maximum Likelihood method and the General Time Reversible model with invariant sites (GTR + I) and 1000 bootstrap replicates. The tree was rooted using CHIKV as the out-group.

### 2.6. GETV MM2021 Sequencing

C6/36 cells were infected with GETV at MOI 0.01 and incubated for 2 days, and culture supernatant was centrifuged at 3000 rpm for 15 min. PEG6000 (Sigma Aldrich, Darmstadt, Germany) was added to culture supernatant to a final concentration of 10%, incubated on a rotor overnight at 4 °C, and then centrifuged at 12,000 rpm for 1 h at 4 °C. The pellet was resuspended in RAV1 lysis buffer from the Nucleospin RNA virus kit (Machery-Nagel, Duren, Germany), and viral RNA was purified as per manufacturer instructions. RNA was sent to Australian Genome Research Facility (AGRF) for Next Generation Sequencing on the MiSeq platform (Illumina, California, USA), and 1,143,407 paired-end reads (150 bp) were generated. Reads were aligned to the GETV MI-110-C2 strain sequence (LC079087.1) using STAR Aligner and the consensus sequence was obtained using Integrative Genomics Viewer (IGV). A complete genomic sequence of the isolate was obtained (GenBank MN849355). Alignment analyses showed a > 99.69% nucleotide identity with the available partial sequence of MM2021 (GenBank AF339484).

### 2.7. Quantitative RT PCR of RRV Infected Feet

Quantitative RT-PCR (qRT-PCR) was performed using iTaq Universal SYBR^®^ Green Supermix (Bio-Rad, California, USA) and RRV-specific NS3 primers: forward 5′-CCTGTYGAGGACGCCGATT-3′ and reverse 5′-CRTACCTACACAGACACGGAACTG-3′ (Thermo Fisher Scientific, Massachusetts, USA) with cycling conditions as described [68], with normalization to the housekeeping gene, RPL13A [69]. Each 20 μL reaction mix contained Supermix (10 μL; Bio-Rad, California, USA), RRV forward and reverse primer (0.3 µM), DNA template (2 μL) and nuclease-free water (6 μL; Sigma-Aldrich, Darmstadt, Germany). Reactions were run using the CFX 96 touch PCR detection system (Bio-Rad, California, USA) and data analyzed using Biorad CFX Real Time Analysis software.

### 2.8. Histology and Immunohistochemistry

Histology, immunohistochemistry and quantitation of stain were undertaken as previously described [42,55,57,70]. Briefly, feet were fixed in 10% formalin, decalcified with EDTA, embedded in paraffin and sections stained with hematoxylin and eosin (H&E; Sigma-Aldrich, Darmstadt, Germany). For immunohistochemistry, sections were stained with anti-CD3 (A0452; Dako, North Sydney, Australia) or anti-F4/80 (AB6640; Abcam, Melbourne, Australia), and detection used ImmPRESS-AP Anti-Rabbit IgG (alkaline phosphatase) Polymer Detection kit (Vector Laboratories, Burlingame, CA, USA) and Warp Red™ Chromogen Kit (Biocare Medical, Pacheco, CA, USA). Slides were scanned using Aperio AT Turbo (Aperio, Vista, CA USA) and analyzed using Aperio ImageScope software (Leica Biosystems, Mt Waverley, Australia) (v10) and the Positive Pixel Count v9 algorithm.

### 2.9. The Alphaviral Vaccines and Vaccination

The multiplication-defective, vaccinia-virus-based, Sementis Copenhagen Vector vaccine encoding the structural polyprotein cassettes of Zika virus and CHIKV (SCV-ZIKA/CHIK) and the control vaccine (SCV-Control) have been described previously [16,28,54] and were kindly provided by Sementis Ltd., Berwick, Australia. The CHIKV sequence was derived from strain 06-021 (GenBank: AM258992.1) obtained from a patient on La Reunion island in 2005 [71].

The formalin-inactivated whole virus JEV/GETV vaccine was purchased from Nisseiken Co. Ltd., Tokyo, Japan and was imported from NIID, Japan under Australian DAWR permit no. 0002721876, which restricted use to class 5 approved arrangement (AA) sites at QIMRB (quarantine facilities). The vaccine was developed using the MI-110 isolate (GenBank: LC079087.1) obtained from an infected horse during the 1978 GETV outbreak at the Miho training center, Ibaraki Prefecture, Japan [31].

The vaccines were administered intramuscularly (i.m.), with the dose split equally into both quadriceps muscles of restrained mice in 50 µL per muscle using an insulin syringe.

### 2.10. Antibody ELISA and Neutralization Assays

IgG responses were determined by standard ELISA using whole alphavirus as antigen and the mean plus 3 SD of naïve serum values as the endpoint as described [24]. Alphaviruses were purified from infected C6/36 cell supernatants by 40% polyethylene glycol precipitation (PEG6000) and ultracentrifugation (~134,000 rcf for 2 h at 4 °C) through a 20% sucrose cushion. Neutralization assays were performed essentially as described [24]. Heat-inactivated (56 °C, 30 min) mouse serum was incubated (in duplicate) with 150 CCID_50_ of alphavirus for an hour at 37 °C before adding to Vero cells (except for ONNV where BHK cells were used) (10^5^ cells/well, 96-well plate). After 5 days, cells were fixed and stained with formaldehyde and crystal violet and the 50% neutralizing titers interpolated from optical density (OD) versus dilution plots.

### 2.11. E1/E2 Contact Residue Visualizations

The figures were generated using Pymol Molecular Graphics System (version 2.3.3) (Schrodinger, NY, USA) using the structure described in Song et al. 2019 [72] (PDB: 6JO8).

### 2.12. Serum, IgM and Ribavirin Treatments of Rag^-/-^ Mice

Anti-CHIKV convalescent serum was collected from 18 mice infected 3–4 weeks previously with CHIKV. The serum was pooled and 200 µL injected intraperitoneally (i.p.) into persistently infected Rag1^-/-^ mice.

Rag2^-/-^ mice persistently infected with RRV_TT_ were treated (i) with an anti-RRV purified IgM monoclonal antibody, 37B2 [73], a control antibody 3G1.1 [74] (400 µg) or PBS (i.p. 400 µL) or (ii) with ribavirin (1-β-D-ribofuranosyl-1H-1,2,4-triazole-3-carboxamide) (Sigma-Aldrich, Darmstadt, Germany) (i.p. 100 mg/kg) or diluent (PBS 100 µL) for five consecutive days. Viremia levels were determined as described above.

### 2.13. Statistics

Statistical analyses of experimental data were performed using IBM SPSS Statistics for Windows, Version 19.0 (IBM Corp., Armonk, NY, USA). The t-test was used when the difference in variances was <4, skewness was >2 and kurtosis was <2. Otherwise, the non-parametric Kolmogorov–Smirnov test was used. Correlation analyses used the non-parametric Spearman’s rank-order correlation.

## 3. Results

### 3.1. Characterisation of RRV_TT_, a Contemporary Human Isolate from an RRV Disease Patient, in Wild-Type Mice

The RRV_T48_ strain has been extensively used in mouse models of RRV musculoskeletal disease [75,76,77]. RRV_T48_ is a mosquito-derived, mouse-adapted isolate, with a largely undocumented passage history [60]. A contemporary clinical RRV isolate was recently obtained from a transfusion case, where the donor had both symptoms of RRV (fatigue and arthralgia) and tested positive for RRV by serology [53]. We sequenced this isolate (RRV_TT_; GenBank accession number: KY302801) with phylogenetic analyses illustrating that it clustered with another human isolate (98% nucleotide identity) but was relatively distant from RRV_T48_ (96% nucleotide identity) (Appendix A), with both conservative and non-conservative amino acid differences in structural and non-structural proteins evident between RRV_T48_ and RRV_TT_ (Appendix A). Nevertheless, the binding of a small panel of monoclonal antibodies was the same (Appendix A), and in vitro replication was largely comparable, (Appendix A) for these two virus isolates.

To the best of our knowledge, no human RRV isolate has been tested for its ability to infect and cause disease in mice. Adult (6–10 week old) female C57BL/6J mice were thus infected s.c. in the hind feet with RRV_T48_ or RRV_TT_ at a dose of 10^4^ CCID_50_. Viremia was significantly higher for RRV_T48_-infected mice compared with RRV_TT_ (Figure 1a, days 1–3), with the age of the mice (6- versus 24-week-old) not having a significant effect on peak viremia (Figure 1a, inset bar graph). Both the viral titers on days 2 and 6 (Figure 1b) and viral RNA levels on days 6 and 30 post-infection (Figure 1c) in the feet were not significantly different for the two RRV isolates. Foot swelling was, however, significantly higher for RRV_TT,_ although it only reached a maximum mean increase of≈12% (Figure 1d), substantially lower than the >60% increase seen in the adult wild-type mouse model of CHIKV arthritis [42,75], perhaps consistent with the substantially lower overall pathogenicity of RRV when compared with CHIKV in humans [4]. The RRV_TT-_ associated swelling, although evident in 6-week-old mice, was absent in 24-week-old mice (Figure 1d, inset bar graph). Age dependence for alphavirus mouse models is well described [78], with the traditional RRV_T48_ mouse model using 3-week-old weanling mice so that an overt musculoskeletal phenotype can be observed [75].

H&E staining of arthritic feet from 6-weeks-old RRV-infected mice day 6 post-infection illustrated the characteristic mononuclear cellular infiltrates [4,79] that were clearly evident in muscle tissues (Figure 1e, black ovals), with some also evident around joint tissues (Figure 1f, black ovals). Quantitation using blue (nuclear)/red (cytoplasmic) staining ratios showed that cellular infiltrates were not significantly different for RRV_T48_ and RRV_TT_ (Figure 1g). (Leukocytes tend to have higher nuclear to cytoplasm ratios than resident cells, so blue/red ratios represents a measure of leucocyte infiltration [56]). Subcutaneous edema was, however, more evident in RRV_TT_-infected mice (Figure 1h, edema indicated by asterisks), perhaps explaining the foot swelling differences seen in Figure 1d.

CD4+ CD3+ T cells and monocytes/macrophages are intimately associated with alphaviral arthritides [5,80]. Immunohistochemistry (IHC) for anti-CD3 on day 6 post-infection illustrated that infiltrating CD3+ T cells were often found in muscle tissues (Figure 1i, black ovals). RRV_TT_ arthritis was associated with more infiltrating CD3+ T cells when compared with RRV_T48_ arthritis (Figure 1j). F4/80+ monocytes/macrophages were often found in subcutaneous connective tissues (Figure 1k) and were not significantly different for the two RRV strains (Figure 1l). Thus, both RRV_TT_ and RRV_T48_ infections of adult wild-type mice show the inflammatory infiltrates characteristic of alphaviral arthritides [4].

RRV_TT_ therefore represents a contemporary (non-mouse-adapted) human isolate of RRV that is suitable for comparison, in adult wild-type mouse models, with other (non-mouse-adapted) human arthritogenic alphavirus isolates.

### 3.2. Protection and Cross-Protection Mediated by Arthritogenic Alphaviruses Infection

Antibody responses directed at E1 and/or E2 proteins are believed to be the primary mediators of protection against arthritogenic alphaviruses [72,81], although other factors may play a minor role [56,82]. The phylogenetic relationships between the E1/E2 amino acid sequences for alphaviruses used herein are shown in Figure 2a.

Using the new human RRV_TT_ isolate and human isolates of CHIKV, ONNV and MAYV (Figure 2a), adult C57BL/6J mice were infected with one alphavirus and were then challenged with the same (homologous) or different (heterologous) alphavirus (Figure 2b). As expected, in all four homologous combinations, complete protection against viremia was observed (Appendix A, middle top graphs, red lines). Complete protection is defined herein as no post-challenge viremia detected on any day in any mouse; (limit of detection 2 log_10_CCID_50_/mL of serum).

As might also be expected amongst members of the same antigenic complex [41], in all heterologous combinations, some level of significant cross-protection was always evident (Figure 2c, see *p*-values). However, for half of the heterologous combinations, cross-protection was partial (Figure 2c, middle column of graphs and third row of graphs). Thus, although all these viruses are clearly antigenically related, complete cross-protection was not universally assured. MAYV infection completely protected against delectable RRV_TT_ viremia, whereas RRV_TT_ infection only partially cross-protected against MAYV viremia (Figure 2c). Although this might be interpreted as some form of one-way cross-protection [83], it should be noted that the “immunizing” viremias were (at their peak) ~100-fold higher for MAYV than for RRV_TT_ (Appendix A).

### 3.3. Conservation of Receptor Contact Residues in E1 and E2

Although antibody-based virus neutralization can involve a range of mechanisms [5,46,85], most neutralizing antibodies target E1/E2, and many block receptor binding [81]. The 48 contact residues between the arthritogenic alphavirus receptor, MXRA8, and the E1/E2 heterotrimeric envelope glyoproteins of CHIKV were recently identified by cryo-electron microscopy [72]. These residues are relatively well conserved between CHIKV and ONNV (77% amino acid identity) but are less well conserved between CHIKV and MAYV (60% amino acid identity) and CHIKV and RRV (48% amino acid identity) (Figure 3a,b). The contact residues are represented on cryo-electron microscopy images, with color-coding (as in Figure 3a) illustrating the levels of conservation (Figure 3b).

Convalescent CHIKV, ONNV, MAYV and RRV_TT_ sera from individual mice were used to determine the neutralization and cross-neutralization titers against CHIKV. When these titers were plotted against the percentage of amino acid identity in the E1/E2 receptor-binding residues, a highly significant relationship became evident (Figure 3c, left graph). Both the significance (*p*-value) and the correlation coefficient (rho) were lower when the same analysis was undertaken using the percentage of amino acid identity for all of the amino acids in E1/E2 (Figure 3c, middle graph), or the percentage of amino acid identity for all of the amino acids encoded by the viral genomes (Figure 3c, right graph). The level of cross-protection mediated by a polyclonal antibody response would thus appear to be closely related to the level of conservation in the E1/E2 receptor-binding residues.

Another key observation here is that reductions in the percentage of amino acid identities resulted in large drops in cross-neutralization titers. For example, an approximate halving of the percentage amino acid identity from 100% to 48% resulted in a ~100 fold drop in cross-neutralization titers (Figure 3c, left graph), likely explaining why only partial cross-protection was frequently observed in Figure 2c.

### 3.4. Cross-Protection Provided by SCV-ZIKA/CHIK Vaccination in Wild-Type Mice

We have previously shown that a single vaccination of adult female C57BL/6J mice with 10^6^ pfu of the SCV-ZIKA/CHIK vaccine i.m. induced CHIKV-specific ELISA and neutralizing antibody responses and provided complete protection against CHIKV viremia after CHIKV challenge [16]. To determine whether such SCV-ZIKA/CHIK vaccination could provide cross-protection against other human arthritogenic alphaviruses in the same antigenic complex, C57BL/6J mice were vaccinated with 10^6^ pfu SCV-ZIKA/CHIK, and antibody responses and protection against ONNV, RRV_TT_ and MAYV infection were determined (Figure 4a). Vaccination with 10^6^ pfu of SCV-ZIKA/CHIK generated ELISA and neutralizing responses specific for both CHIKV and ONNV, although ELISA responses were slightly lower (1.6 fold) (Figure 4b), and neutralization titers were ~10-fold lower for ONNV when compared with CHIKV (Figure 4c). Nevertheless, the mean cross-neutralization titer against ONNV of 40.2 + SE 9.6 (Figure 4c) was sufficient for complete cross-protection against detectable viremia post-ONNV challenge in this model (Figure 4d, red arrow). Although SCV-ZIKA/CHIK vaccination did induce a mean reciprocal endpoint anti-MAYV ELISA titer of 111 + SE 30 (Figure 4e), no cross-neutralization titers were detected (Figure 4f), and no cross-protection was observed (Figure 4g). No cross-reactive (Figure 4h) or cross-neutralizing (Figure 4i) antibody responses against RRV_TT_ were observed, and a single vaccination with 10^6^ pfu of SCV-ZIKA/CHIK was unable to provide any cross-protection against viremia (Figure 4j).

To determine whether increased SCV-ZIKA/CHIK vaccination dose and/or boosting might provide cross-protection against RRV_TT_, mice were vaccinated twice with 10^6^ pfu SCV-ZIKA/CHIK, once with 10^7^ pfu SCV-ZIKA/CHIK (Appendix A) or twice with 10^7^ pfu SCV-ZIKA/CHIK (Figure 4k). Only the latter dose and schedule induced RRV-specific antibody responses detectable by ELISA, although mean levels were ~15-fold lower than CHIKV-specific ELISA titers (Figure 4l, *p* = 0.005). No cross-neutralizing antibody responses were detected against RRV_TT_ (Figure 4m); nevertheless, partial cross-protection against RRV_TT_ viremia after RRV_TT_ challenge was evident (Figure 4n), with 3 out of 6 mice showing no detectable viremias. Thus even when SCV-ZIKA/CHIK was given at a 10-fold higher dose and twice, only 50% of the mice were fully protected against a detectable RRV_TT_ viremia.

### 3.5. Cross-Protection Against RRV_TT_ Mediated by the JEV/GETV Vaccine in Wild-Type Mice

Of the arthritogenic alphaviruses described herein, GETV is the most closely related to RRV (Figure 3a). A formalin-inactivated JEV/GETV vaccine is available in Japan for use in horses (Nisseiken, Tokyo, Japan) [31,32,33] and currently represents the only arthritogenic alphavirus for which a vaccine is commercially available. RRV also infects horses [37,38], but no commercial RRV vaccine is currently available for veterinary or human use.

We determined that the JEV/GETV vaccine contains 172 µg/mL of protein using a standard Bradford protein assay. The equine 3 mL dose thus represents ~500 µg of protein, with two i.m. doses separated by 1 month recommended for horses by the manufacturer. To explore whether the JEV/GETV vaccine could provide cross-protection against RRV, mice were vaccinated once i.m. with 10 µg of the JEV/GETV vaccine (Figure 5a); assuming 50% of this vaccine is GETV, this represents a vaccination dose of ~5 µg of GETV proteins. An experimental formalin-inactivated RRV vaccine at a single dose of 2.5 µg was previously shown to induce mean reciprocal neutralizing titers of ~30 and provided protection against viremia in 93% of CD-1 mice [48]. Reciprocal 50% neutralizing antibody titers for RRV of >10 were deemed a conservative estimate of protection in a human vaccine trial [22].

Mice vaccinated once with 10 µg of JEV/GETV vaccine generated a mean reciprocal anti-GETV ELISA antibody titer of 4141 + 1365, with the mean reciprocal anti-RRV_TT_ ELISA titers ~100 fold lower at 37.7 + SE 23.1 (Figure 5b). The mean reciprocal anti-GETV neutralizing titer was 35 + SE 20, with anti-RRV_TT_ cross-neutralizing titers below the limit of detection (Figure 5c). PBS vaccination induced no detectable antibody responses (Figure 5b,c). JEV/GETV vaccination at this dose produced no detectable protection against RRV_TT_ challenge (Figure 5d). Mice vaccinated twice with 20 µg of JEV/GETV vaccine (Figure 5e) generated a mean reciprocal anti-GETV ELISA antibody titer of 2578 + SE 854, with the mean anti-RRV_TT_ ELISA titers 14-fold lower at 179 + SE 41 (Figure 5f). The mean reciprocal anti-GETV neutralizing titer was 38.7 + SE 16, and the anti-RRV_TT_ cross-neutralizing titers were again below detection (Figure 5g). The increased dose and second vaccination (Figure 5a vs. Figure 5e) thus provided no significant increase in anti-GETV responses (Figure 6b vs. Figure 6f and Figure 6c vs. Figure 6g) or anti-RRV_TT_ neutralizing antibody responses (Figure 5c vs. Figure 5g) but did increase anti-RRV ELISA titers approximately 5-fold (Figure 5f vs. Figure 5b; 179/37.7 = 4.75). Following challenge with RRV_TT_, significant, but only partial, protection against viremia was observed (Figure 5h). (As we do not as yet have a mouse model of GETV, we were unable to verify that the commercial JEV/GETV vaccine can protect against GETV in mice.)

The E1/E2 receptor binding residues are relatively well conserved for both RRV_TT_ and GETV_MI-110_ (the vaccine strain), and RRV_TT_ and GETV_MM2021_ (the GETV isolate used for the ELISA and neutralization assays) (Figure 5i). Given the 75% amino acid identity between RRV_TT_ and GETV_MI-110_ (Figure 5i), the poor level of cross-protection (Figure 5d,h) might be viewed as somewhat unexpected. However, the JEV/GETV vaccine is formalin-fixed, a process known to reduce immunogenicity [87], potentially by altering the antigenic structure [88]. Formalin irreversibly modifies certain epitopes, particularly those containing lysine (K) and (to a lesser extent) tryptophan (W) residues [89]. Induction of cross-reactive antibodies that are reliant on conserved epitopes that (for instance) contain E1 K^130^ and/or W^89^ (Figure 5i, underlined) may thus be compromised.

### 3.6. Cross-Protection Against RRV_TT_ after SCV-ZIKA/CHIK Vaccination in IRF3/7^-/-^ Mice

Mice defective in type I interferon (IFN) responses, such as IFN alpha-receptor-deficient (IFNAR^-/-^ or A129) mice, have been widely utilized to evaluate alphaviral vaccines and antibodies, with protection against mortality often used as an indicator of efficacy [45,52,90,91,92]. Such mice have also been used to demonstrate cross-protection against ONNV by a CHIKV vaccine [47]. Herein we used Interferon Response Factor 3- and 7-deficient (IRF3/7^-/-^) mice rather than IFNAR^-/-^ mice, as severe disease has a slower onset [55], which facilitates compliance with ethically defined endpoints for euthanasia, as monitoring of animal well-being can be reduced to twice daily. The behavior of CHIKV in IRF3/7^-/-^ and IFNAR^-/-^ mice is otherwise very similar; both IFNAR^-/-^ and IRF3/7^-/-^ mice are unable to mount protective type I IFN responses and generally show a lethal phenotype associated with hemorrhagic shock after CHIKV infection [55,93]. The latter represents a severe disease manifestation occasionally seen in CHIKV and MAYV patients [5]. To our knowledge, RRV infection of IRF3/7^-/-^ mice has not previously been reported, with infection (as might be expected [55]) resulting in high viremia, foot swelling (edema) and mortality (Appendix A). Subcutaneous monocyte/macrophage infiltration was also illustrated by immunohistochemistry (Appendix A); a feature not previously reported for alphavirus infections in type I IFN response defective mice.

To evaluate cross-protection in type I interferon-response-deficient mice, IRF3/7^-/-^ mice were vaccinated once with 10^6^ pfu SCV-ZIKA/CHIK and were then challenged with RRV_TT_ (Figure 6a). Vaccination induced CHIKV-specific ELISA and neutralizing antibody responses, but anti-RRV_TT_ antibody responses were below the level of detection (Figure 6b,c). After RRV_TT_ challenge, both SCV-ZIKA/CHIK and SCV-control vaccinated mice developed high viremias; although on days 4 and 5, post-challenge SCV-ZIKA/CHIK-vaccinated mice showed significant 0.8-1.4 log lower viremia titers (Figure 6d). After RRV_TT_ challenge, both SCV-ZIKA/CHIK and SCV-control vaccinated mice showed similar levels of foot swelling (Figure 6e); images of typical foot swelling in control mice are shown in Figure 6f. Mice were also monitored for a series of disease signs (1 = mild, 2 = moderate, 3 = severe); any animal scoring 2 in two or more disease parameters (or 3 in any one parameter) was deemed to have reached an ethically defined endpoint requiring euthanasia. No SCV-ZIKA/CHIK-vaccinated animals reached this criterion, whereas all SCV-control vaccinated mice scored 2 for Joint swelling and 2 for Posture on day 6 and were euthanized (Figure 6g). Presented as a Kaplan–Meier survival plot, SCV-ZIKA/CHIK vaccinated animals showed significantly better survival than SCV-control vaccinated mice (Figure 6h).

SCV-ZIKA/CHIK vaccination of IRF3/7^-/-^ mice thus provided no cross-protection against RRV_TT_ foot swelling, marginal cross-protection against viremia and disease, but complete cross-protection against mortality. The same vaccination also provided no protection against RRV_TT_ viremia in wild-type mice (Figure 4j). Survival in the type I interferon-deficient mouse model is thus a poor indicator of (or poor surrogate marker for) cross-protection against infection and arthritic disease, with low-level cross-reactive anamnestic responses perhaps only just able to generate sufficient immune responses in time to prevent mortality.

### 3.7. Serological Protection and Cross-Protection Evaluated in Rag^-/-^ Mice

Anti-alphaviral IgG responses are well described as mediators of protection [5,7,46,81], with T cells perhaps playing a minor role [56,82]. The alphavirus cross-protection literature has traditionally implicated T cells [51,94,95,96,97,98], although antibody responses have also been implicated [42,50] and cross-reactive and cross-neutralizing antibodies are well described [43,44,45,99].

Rag^-/-^ mice have no B or T cell responses and are unable to clear CHIKV or RRV infections resulting in long-term steady-state viremias [56,82]. Rag1^-/-^ mice were infected with CHIKV, ONNV, MAYV or RRV_TT_ (*n* = 4 per group), and after the persistent viremias were established, mice were treated on day 10 with pooled CHIKV convalescent sera (Figure 7a). The sera were pooled from 18 CHIKV-infected mice and had a reciprocal CHIKV IgG ELISA endpoint titer of ~50,000. With the exception of RRV_TT_, all viremia levels dropped to levels below detection within a day, with the MAYV viremia reappearing by day 20 in three mice and by day 30 in the remaining mouse (Figure 7b; Appendix A). The ONNV and CHIKV viremia levels remained depressed for >47 days and were not significantly different from each other (on days 34 and 39) (Figure 7b). The results are consistent with the ability of SCV-ZIKA/CHIK vaccination to protect against CHIKV [16] and cross-protect against ONNV, but not MAYV or RRV (Figure 4). This pattern (Figure 7b) also correlates well with amino acid identity in receptor contact residues [81] shown in Figure 3b (Appendix A); with the cross-protection activity of anti-CHIKV sera against persistent viremia, best for ONNV (77% identity), less for MAYV (60% identity) and least for RRV_TT_ (48% identity) (Figure 7b). Taken together, these data implicate cross-neutralizing IgG responses as playing an important role in mediating cross-protection. (These data also illustrate the utility of Rag^-/-^ mice for evaluating anti-viral treatments for arthritogenic alphaviruses, with further examples shown in Appendix A.)

## 4. Discussion

We illustrate herein that some level of cross-protection, mediated by CHIKV, ONNV, MAYV and RRV infections, was universal in adult wild-type mice, as might be envisaged given these viruses all belong to the Semliki Forest virus antigenic complex. However, in half the combinations, cross-protection was only partial (Figure 2c, Appendix A). The live attenuated recombinant poxvirus-based SCV-ZIKA/CHIK vaccine was able to mediate complete cross-protection against ONNV challenge but was unable to provide such protection against MAYV or RRV (Figure 4). The formalin-inactivated JEV/GETV vaccine was unable to provide complete cross-protection against RRV challenge, despite the high E1/E2 sequence identities between these two viruses (Figure 2a and Figure 5i). Both infection-mediated and vaccine-mediated cross-protection were thus often partial or absent.

Although SCV-ZIKA/CHIK vaccination provided effective cross-protection against ONNV challenge (Figure 4d), the mean neutralizing antibody response was ~10-fold lower for ONNV than for CHIKV (Figure 4c). In addition, after SCV-ZIKA/CHIK vaccination, the ratio of neutralizing titers over ELISA titers was significantly lower for ONNV than for CHIKV (Appendix A, *p* < 0.001), further illustrating that the SCV-ZIKA/CHIK vaccine is less ideal for ONNV than it is for CHIKV. In humans, the minimum vaccine dose to achieve protection against its designated target is generally used to avoid excessive side effects such as reactogenicity [100] and to reduce the cost of goods. In humans, reciprocal neutralization titers of >10 have been deemed to represent a conservative estimate of the protective titer for RRV [22] and for CHIKV [101,102]; a contention not inconsistent with the mouse data presented herein (Appendix A). Thus, if the SCV-ZIKA/CHIK vaccine dose was reduced in humans to provide such levels of anti-CHIKV neutralization, the lower cross-neutralization titers against ONNV may provide suboptimal cross-protection against ONNV. Testing in clinical trials and ultimate licensing of a vaccine that requires higher or more vaccine doses in order to also provide cross-protection might present regulatory hurdles when the technology (if not the market) likely exists to generate an appropriately targeted vaccine. Vaccine manufacturers are also likely to focus on the primary target and not complicate registration and licensing by risking increased side effects (and countenancing increased cost of goods) in an attempt to capture an additional, economically less viable, cross-protection market.

Despite the high level of sequence identity between GETV and RRV, JEV/GETV vaccination was unable to induce anti-RRV cross-neutralizing antibody responses (Figure 5c,g) or to provide complete protection against RRV_TT_ challenge (Figure 5d,h). Anti-RRV_TT_ ELISA responses were also 100- and 14-fold lower than the anti-GETV ELISA titers (Figure 5b,f, respectively), again suggesting that cross-protection against RRV would require a higher dose of the JEV/GETV vaccine than that needed for protection against GETV. Even the anti-GETV neutralizing responses generated by the GETV vaccine were relatively poor, perhaps suggesting that a very high dose or multiple vaccinations would be needed before cross-neutralizing responses to RRV are seen. The suboptimal induction of neutralizing antibody responses by a formalin-fixed vaccine [87,88,89] is perhaps supported herein by the observation that the ratios of neutralizing titers over ELISA titers were significantly higher for SCV-ZIKA/CHIK than for JEV/GETV vaccinations (Appendix A, *p* < 0.0001). Generating cross-neutralizing responses with a formalin-fixed vaccine may thus represent a particularly difficult hurdle.

Herein, protection against viremia was used as the readout, rather than protection against disease, which primarily manifests in humans as acute and often chronic polyarthralgia/polyarthritis [4,5]. After injection of 10^4^ CCID_50_ of CHIKV s.c. into the foot of adult wild-type mice, the arthritic foot swelling reached a maximum mean ~60%–80% increase in foot height x width [56,103]. For RRV_TT_, such foot swelling only reached ~12% in 6-week-old mice and was essentially below detection in 24-week-old mice (Figure 1d). Although a higher dose (10^5^ or 10^6^ PFU) and different strain of MAYV (BeH407) has been reported to induce foot swelling comparable to that seen for CHIKV after s.c. injection into the feet [104], no significant foot swelling was seen herein in control mice challenged with 10^4^ CCID_50_ MAYV (or ONNV). These alphaviruses thus show distinct differences in their propensity to induce foot swelling in these wild-type mouse models, making cross-protection against arthritis difficult to standardize. No mouse models have so far been developed that measure polyarthralgia. The adult wild-type CHIKV mouse model recapitulates many aspects of human disease [42,57], with recent RNA-Seq comparisons further illustrating that this mouse model shares many inflammatory processes with those seen in humans [105,106]. However, such comparisons are not yet available for ONNV, MAYV or RRV. Further development and characterization of alphavirus mouse models would thus be required before one could formally test the inherent assumption herein that suppression of viremia correlates with suppression of rheumatic disease.

Overall the data presented herein does not support the concept of effective mediation of cross-protection by arthritogenic alphavirus vaccines. A vaccine capable of protecting against multiple arthritogenic alphaviruses would likely require a polyvalent vaccine [54] or a mixture of individual vaccines, as was recently described for the three major encephalitic alphaviruses [107]. However, the current small market size and low mortality rates for RRV, ONNV and MAYV would likely make such endeavors commercially unattractive. New outbreaks may clearly change this contention [10,14]; until then, mosquito control measures are likely to remain at the forefront of public health measures against these latter alphaviruses [108].

## 5. Conclusions

A CHIKV vaccine is deemed commercially viable, and several vaccine candidates are progressing into, and through, human clinical trials. Although CHIKV-vaccine or CHIKV-infection provided a level of cross-protection against infection by ONNV, MAYV and RRV, cross-protection and/or cross-neutralizing antibody responses were significantly lower than protection and/or neutralizing antibody responses against CHIKV. Increasing CHIKV vaccine dosing and/or scheduling to try and capture the much smaller cross-protection markets is unlikely to be attractive to regulators or vaccine manufacturers.

## Figures and Tables

**Figure 1 vaccines-08-00209-f001:**
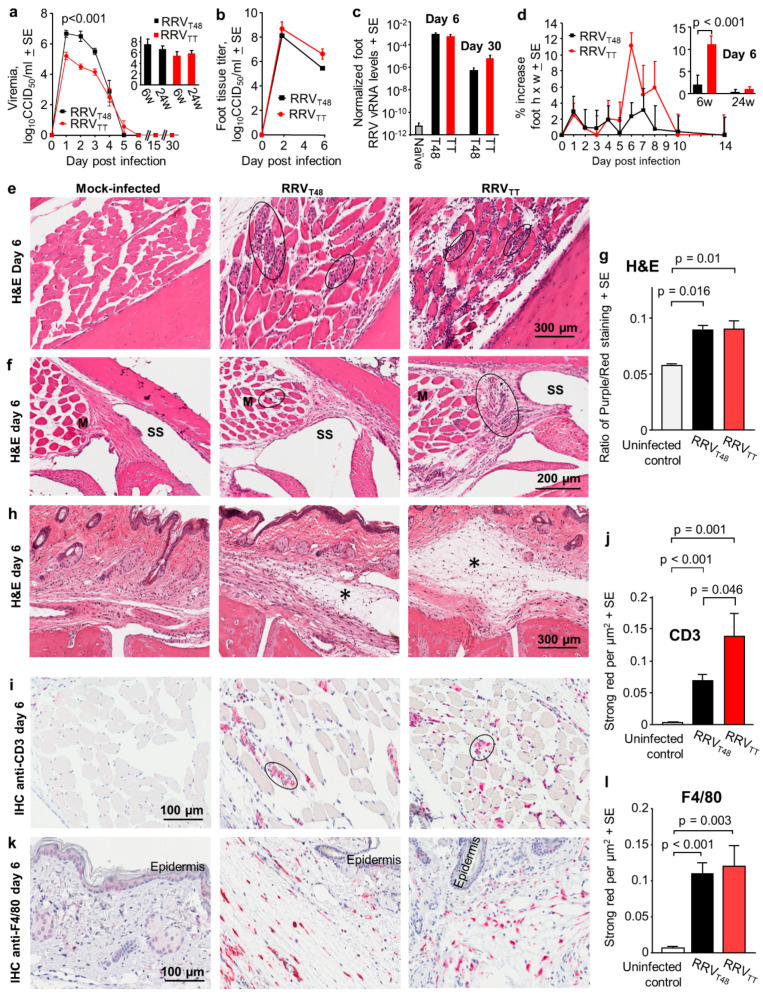
Ross River viruses (RRVs) RRV_TT_ and RRV_T48_ arthritis in adult wild-type mice. (**a**) Viremia in adult C57BL/6J female mice (6- and 24-weeks-old) infected with 4 log_10_CCID_50_ of RRV_TT_ or RRV_T48_. RRV_T48_ viremia was significantly higher on days 1, 2 and 3 (*p* < 0.001); statistics by Kolmogorov–Smirnov tests (*n* = 6–12 mice per time point and data were derived from 3 independent experiments). Bar graph insert shows mean peak viremias for mice infected at 6 or 24 weeks of age; (**b**) Viral titers in feet of C57BL/6J mice infected as in (**a**) (*n* = 6 feet from 3 mice per time point); (**c**) Quantitative RT-PCR of feet from RRV infected mice (taken days 6 and 30 post-infection) using RRV NS3 primers normalized to RPL13a (*n* = 3 feet from 3 mice per group). Naïve represents uninfected mice; (**d**) Percentage increase in foot height x width (relative to day 0) for 6-week-old mice infected as in (**a**), with *n* = 12 feet from 6 mice per group per time point. Inserted bar graph shows the mean peak foot swelling on day 6 post-infection for mice aged 6 weeks (*n* = 12 feet from 6 mice) or 24 weeks (*n* = 6 feet from 3 mice) at time of infection. Differences between strains only reached significance on day 6 (see inserted bar graph; t-test); (**e**) H&E staining of muscle tissues in foot sections from 6-week-old C57BL/6J female mice infected as in (a). Black ovals indicate some of the areas containing inflammatory infiltrates. (*); (**f**) H&E staining as in (**e**) showing inflammatory infiltrates near joint tissues; M = muscle; SS = synovial space; (**g**) Ratio of nuclear (blue/dark purple) to non-nuclear (red) staining of H&E stained foot sections (a measure of leukocyte infiltration). Data from 4–6 feet from 3–4 mice per group, with 2–3 sections scanned per foot and values averaged to produce one value for each foot. Statistics by Kolmogorov–Smirnov tests (as difference in variance were 112 and 26, respectively); (**h**) H&E staining as in (**e**) showing areas of subcutaneous edema (*); (**i**) Immunohistochemical (IHC) staining of sections described in (**e**), with anti-CD3 antibody (primarily T cells) showing positive staining cells in muscle tissues (examples indicated with black ovals); (**j**) Aperio Positive Pixel Count Determination of anti-CD3 staining; strong positive red pixels per µm^2^ in whole foot sections (*n* = 6–12 feet per sample, with 3 sections per foot). Statistics by Kolmogorov–Smirnov tests (as difference in variance were 225 and 3281, respectively); (**k**) IHC staining using anti-F4/80 (monocytes/macrophages) of section described in (**h**), illustrating positive cells in subcutaneous tissues (the epidermis is indicated); (**l**) As for (**j**) except for anti-F4/80 staining. Statistics by Kolmogorov–Smirnov tests (as difference in variance were 70 and 142, respectively).

**Figure 2 vaccines-08-00209-f002:**
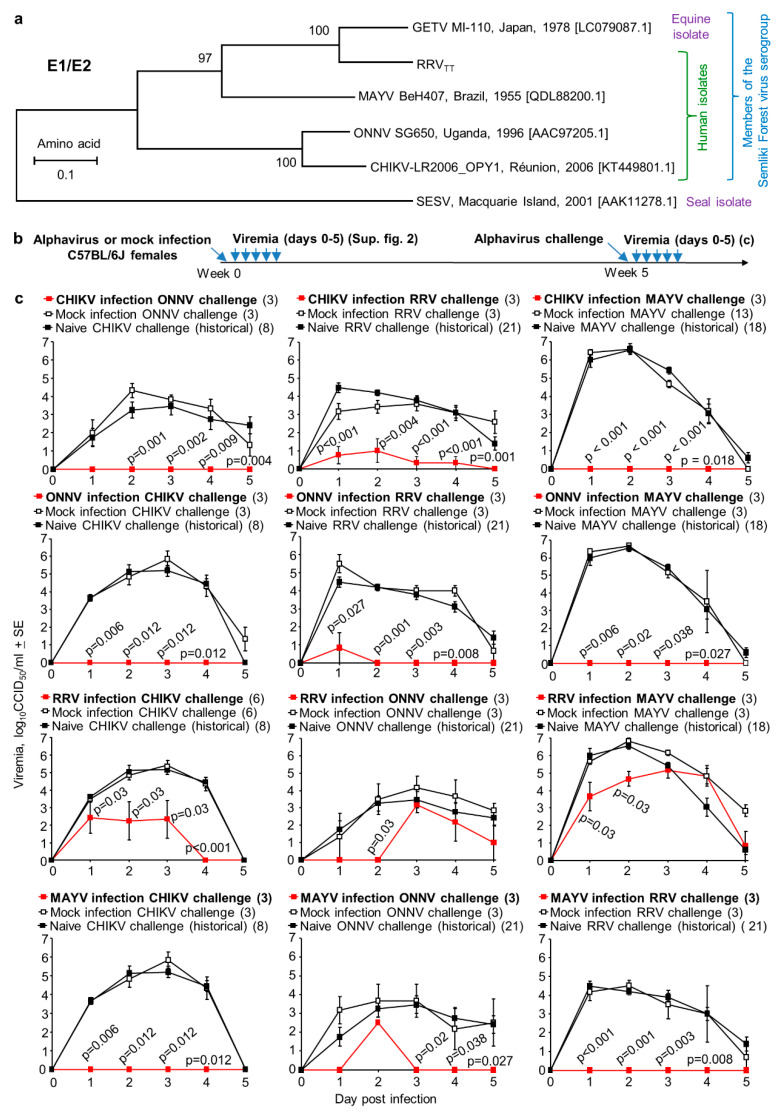
Cross-protection by infection. (**a**) Phylogenetic trees of E1/E2 amino acid sequences (as in Appendix A of arthritogenic alphaviruses used herein, with Southern Elephant Seal virus (SESV) [84] used as an outgroup; (**b**) Timeline of infection, challenge and viremia determinations; (**c**) Heterologous cross-protection. Black lines represent mock-infected or naïve mice challenged with the indicated virus; red lines represent viremia after infection and heterologous challenge. Numbers in brackets represent mouse numbers. Statistics by Kolmogorov–Smirnov tests taking Mock (white fill) and Naïve (black fill) together, and comparing with heterologous challenge group (red).

**Figure 3 vaccines-08-00209-f003:**
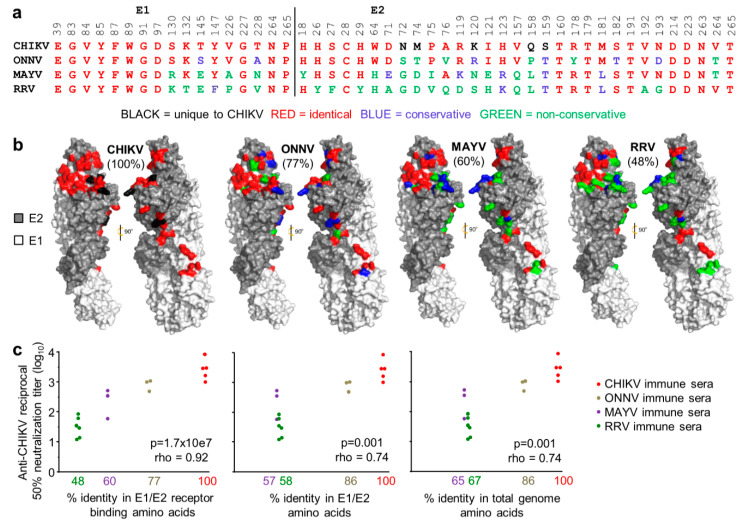
Conservation of receptor contact residues amongst the arthritogenic alphaviruses. (**a**) The 48 residues on the E1/E2 Chikungunya virus (CHIKV) viral spike glycoproteins reported to make contact with the arthritogenic alphavirus receptor, MXRA8 [72] are aligned against the corresponding residues in o’nyong nyong virus (ONNV), Mayaro virus (MAYV) and RRV. Conservative and non-conservative substitutions are defined as described [86]. The complete sequence for ONNV IMTSSA 2004 was not available so residues E1 39-84 and E2 18-123 were derived from the closely related ONNV SG650 [62]; (**b**) The crystal structure of CHIKV E1/E2, with the receptor contact residues colored as in (**a**) for each of the indicated alphaviruses. Percentages indicate the percentage of identical contact residues relative to CHIKV; (**c**) Wild-type C57BL/6J mice were infected with CHIKV, ONNV MAYV or RRV. After 5 weeks, sera from each mouse was tested for its CHIKV neutralization titer. The neutralization titers were plotted against the % identity in E1/E2 receptor binding amino acids (left), the % identity in E1/E2 amino acids (middle) and % amino acid identity in the total genome (right). Statistics by Spearman’s correlations, *p* and rho indicated.

**Figure 4 vaccines-08-00209-f004:**
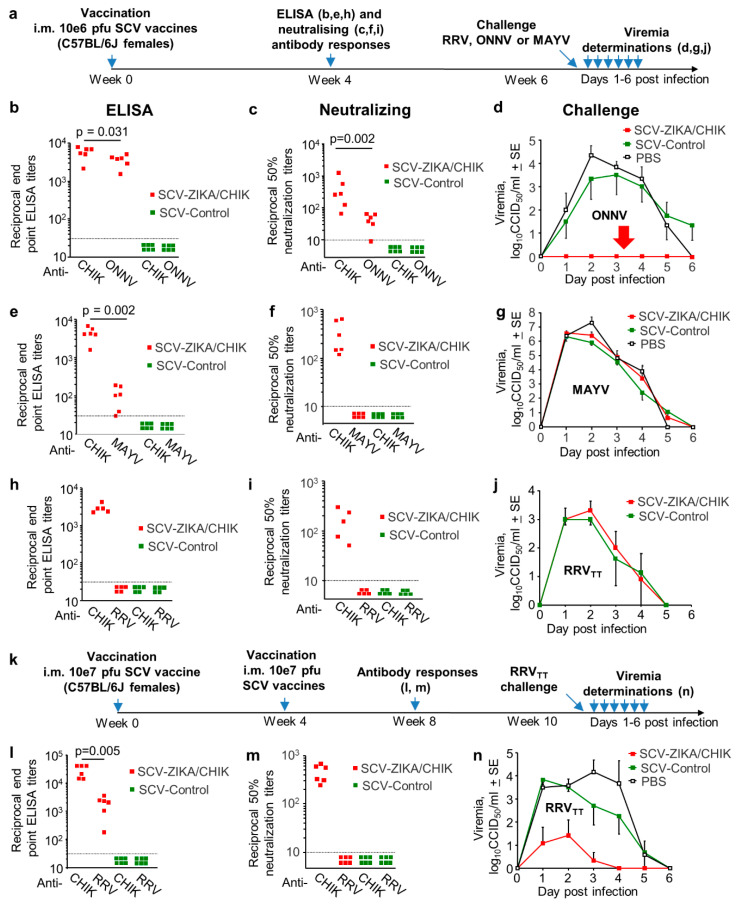
Cross-protection mediated by Sementis Copenhagen Vector (SCV)-ZIKA/CHIK vaccinations in wild-type mice. (**a**) Timeline of vaccination with a single dose of 10^6^ pfu of SCV vaccines, antibody response determinations and challenge; (**b**) CHIKV and ONNV endpoint IgG ELISA titers four weeks post-vaccination with SCV-ZIKA/CHIK or SCV-Control (limit of detection 1 in 30); (**c**) CHIKV and ONNV 50% neutralization titers four weeks post-vaccination with SCV-ZIKA/CHIK or SCV-Control (limit of detection 1 in 10); (**d**) ONNV viremia post-challenge with ONNV in mice vaccinated with SCV-ZIKA/CHIK or SCV-Control or PBS (*n* = 6 mice per group). (Limit of detection for each mouse 2 log_10_CCID_50_/mL of serum.); (**e**) CHIKV and MAYV endpoint IgG ELISA titers four weeks post-vaccination with SCV-ZIKA/CHIK or SCV-Control; (**f**) CHIKV and MAYV 50% neutralization titers four weeks post-vaccination with SCV-ZIKA/CHIK or SCV-Control; (**g**) MAYV viremia post-challenge with MAYV in mice vaccinated with SCV-ZIKA/CHIK or SCV-Control or PBS (*n* = 6 mice per group); (**h**) CHIKV and RRV endpoint IgG ELISA titers four weeks post-vaccination with SCV-ZIKA/CHIK or SCV-Control; (**i**) CHIKV and RRV_TT_ 50% neutralization titers four weeks post-vaccination with SCV-ZIKA/CHIK or SCV-Control; (**j**) RRV_TT_ viremia post-challenge with RRV_TT_ in mice vaccinated with SCV-ZIKA/CHIK or SCV-Control (*n* = 6 mice per group); (**k**) Timeline of 2 vaccinations with 10^7^ pfu of SCV vaccines, antibody response determinations and RRV_TT_ challenge; (**l**) CHIKV and RRV endpoint IgG ELISA titers after 2 vaccinations with SCV-ZIKA/CHIK or SCV-Control; (**m**) CHIKV and RRV_TT_ 50% neutralization titers after 2 vaccinations with SCV-ZIKA/CHIK or SCV-Control; (**n**) RRV_TT_ viremia post-challenge with RRV_TT_ in mice vaccinated twice with SCV-ZIKA/CHIK or SCV-Control or PBS (*n* = 6 mice per group). Viremias were not significantly different for any individual day; taking all viremia data for days 1–4 together (*n* = 24 per group), the SCV-ZIKA/CHIK vaccinated mice had a significantly lower viremia than the SCV-Control vaccinated mice (*p* < 0.001, Kolmogorov–Smirnov test).

**Figure 5 vaccines-08-00209-f005:**
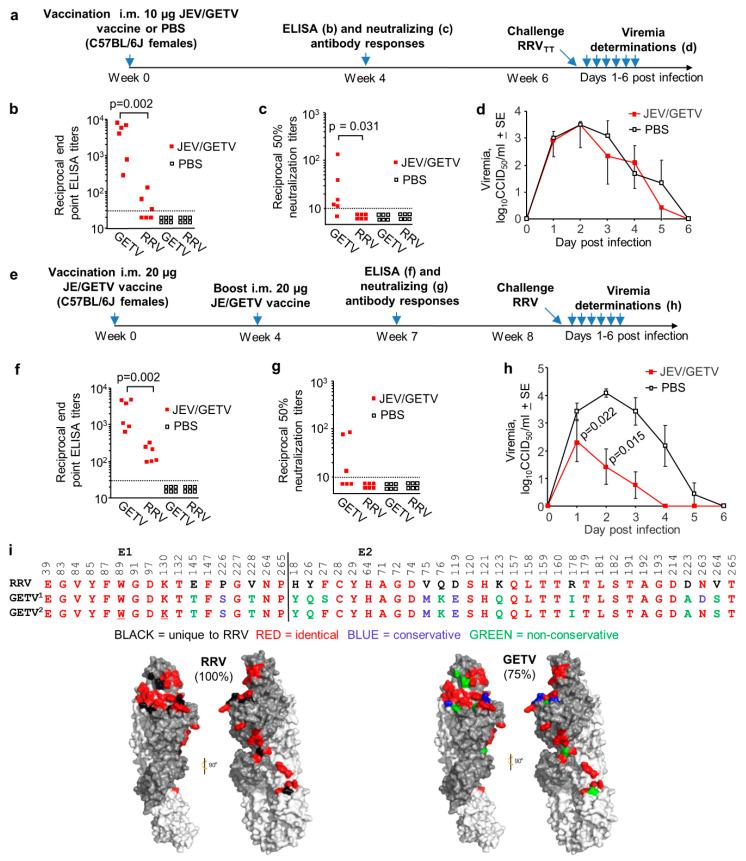
Cross-protection against RRV mediated by the Japanese Encephalitis (JEV)/ Getah virus (GETV) vaccine. (**a**) Timeline of the single 10 µg vaccination with JEV/GETV vaccine (the GETV vaccine strain is MI-110) or PBS, antibody measurements and RRV_TT_ challenge; (**b**) GETV_MM2021_ and RRV_TT_ endpoint IgG ELISA titers after 1 vaccination with JEV/GETV vaccine or PBS (limit of detection 1 in 30); (**c**) GETV_MM2021_ and RRV_TT_ 50% neutralization titers after 1 vaccination with JEV/GETV vaccine or PBS (limit of detection 1 in 10); (**d**) RRV_TT_ viremia post-challenge with RRV_TT_ in mice vaccinated once with JEV/GETV vaccine or PBS (*n* = 6 mice per group). (Limit of detection for each mouse 2 log_10_CCID_50_/mL.); (**e**) Timeline of two 20 µg vaccinations with JEV/GETV vaccine or PBS, antibody measurements and RRV_TT_ challenge; (**f**) GETV and RRV_TT_ endpoint IgG ELISA titers after 2 vaccinations with JEV/GETV or PBS (limit of detection 1 in 30); (**g**) GETV_MM2021_ and RRV_TT_ 50% neutralization titers after 2 vaccinations with JEV/GETV vaccine or PBS (limit of detection 1 in 10); (**h**) RRV_TT_ viremia post-challenge with RRV_TT_ in mice vaccinated twice with JEV/GETV vaccine or PBS (*n* = 6 mice per group). Statistics by Kolmogorov–Smirnov tests; (**i**) As for Figure 3a,b the 48 E1/E2-MXRA8 contact residues (based on CHIKV) are aligned for RRV_TT_ and GETV_MI-110_ (GETV^1^, vaccine strain) and GETV_MM2021_ (GETV^2^) E1/E2, with GETV_MI-110_ residues colored in the cryo-EM structures of E1/E2. GETV_MI-110_ shows 75% amino acid identity in these 48 amino acids with RRV_TT_. The conserved, but formalin-fixation-sensitive, amino acids K and W are underlined.

**Figure 6 vaccines-08-00209-f006:**
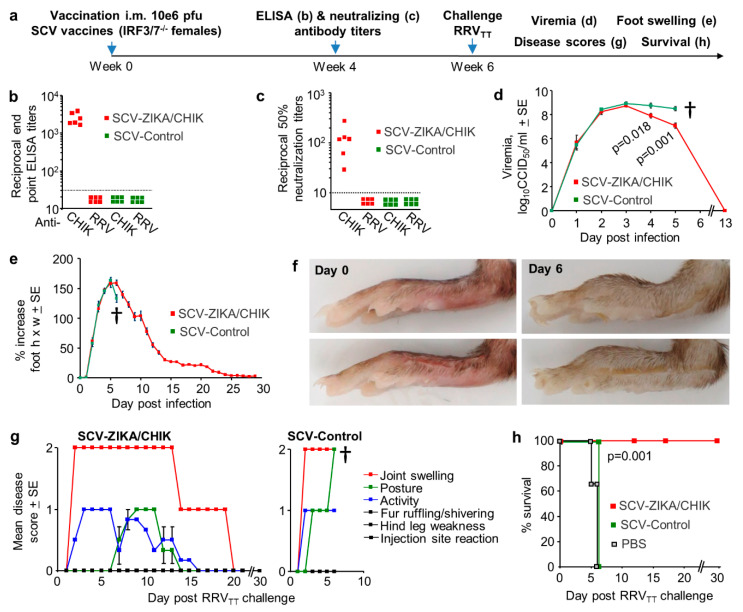
Cross-protection against RRV_TT_ mediated by SCV-ZIKA/CHIK vaccinations in adult IRF3/7^-/-^ mice. (**a**) Timeline of vaccination with a single dose of 10^6^ pfu SCV-ZIKA/CHIK or SCV-Control, antibody measurements and RRV_TT_ challenge in adult IRF3/7^-/-^ mice; (**b**) CHIKV and RRV_TT_ endpoint IgG ELISA titers after vaccination with SCV-ZIKA/CHIK or SCV-Control (limit of detection 1 in 30); (**c**) CHIKV and RRV_TT_ 50% neutralization titers after vaccination with SCV-ZIKA/CHIK or SCV-Control (limit of detection 1 in 10); (**d**) Mean viremias post-challenge with RRV_TT_ (*n* = 6 mice per group). † mice were euthanized. Statistics by t-tests; (**e**) Mean percentage increase in foot swelling (height x width) relative to day 0 for each foot (*n* = 12 feet from 6 mice per group); (**f**) Pictures showing examples of feet on day 0 and day 6 post-infection, illustrating swelling on day 6 in the control group; (**g**) Mice were scored for 1 = mild, 2 = moderate, 3 = severe for the indicated 6 disease manifestations. The mean scores for each disease manifestation for 6 mice per group are shown for SCV-ZIKA/CHIK (left) and SCV-Control vaccinated mice (right); (**h**) Kaplan–Meier survival curves for mice described in (**g**), with mice scoring 2 or more for two disease manifestations deemed to have reached ethically defined endpoints requiring euthanasia. An additional group (*n* = 3) were mock-vaccinated with PBS. Statistics by log rank test.

**Figure 7 vaccines-08-00209-f007:**
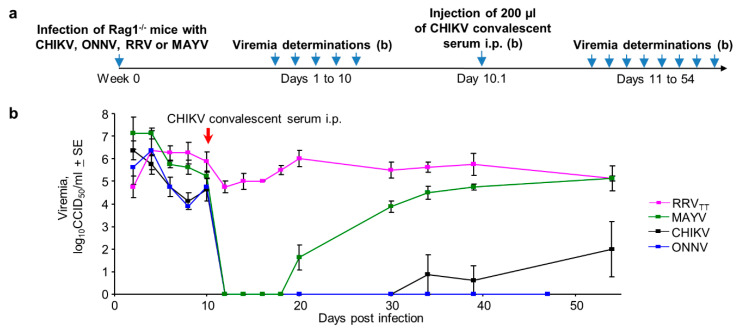
Antibody-mediated cross-protection in Rag^-/-^ mice. (**a**) Timeline of infections, viremia determinations and injection of convalescent CHIKV mouse serum in Rag1^-/-^ mice; (**b**) Viremias for the indicated alphaviruses before and after injection of anti-CHIKV convalescent serum (*n* = 4 mice per group). (Limit of detection for each mouse 2 log_10_CCID_50_/mL.) For all reductions to undetectable viremia levels for each day relative to day 10, *p* = 0.037 by Kolmogorov–Smirnov tests (as differences in variance were infinite).

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
