# Peer review of "Arthritogenic Alphavirus Vaccines: Serogrouping Versus Cross-Protection in Mouse Models"

_vaccines, 2020, doi:10.3390/vaccines8020209_

Round 1
Reviewer 1 Report
The manuscript “Arthritogenic alphavirus vaccines: serogrouping versus cross-protection in mouse models” by Nguyen et al., addresses the concept of universal vaccine against viruses that causes rheumatic manifestations. The manuscript is very clearly presented and very comprehensive and merits acceptance in the journal Vaccines.
The authors have taken a systematic approach to determine homologous and heterologous protection. All the experiments are clearly presented and proper controls are used in the study. Initially authors establish the infection of RRVTT, a human isolate in mice model in comparison with standard RRV strain. It is interesting to find the human isolate produce arthritis with CD3+ T-cell infiltration resulting in significant swelling and demonstrated with IHC staining.
The authors with series of experiments, demonstrate cross-protection after infection with heterologous virus, vaccines and by convalescent sera. The results are clearly presented, and the interpretation is adequate to demonstrate direct correlation between percent identity vs protection. The manuscript characterizes the important aspect of developing a universal vaccine. The results suggest the requirement of multiple vaccines to offer protection.
Based on the clear presentation and interpretation of the results, I strongly recommend acceptance of this article.
Minor comments :
Figure 2 : While RRV infection followed by challenge with MAYV challenge detects viremia, the opposite, MAYV infection followed by RRV challenge completely prevents viremia.
Figure 5 : Does JE/GETV vaccine provides complete protection against GETV virus ? Some of the vaccinated mice (c) and (g) do not develop higher neutralization titers.
Is it possible to use create chimeric/mosaic proteins to generate broad protection as observed with viruses like influenza ?
Author Response
Responses to Reviewer 1 Comments
Point 1: Figure 2: While RRV infection followed by challenge with MAYV challenge detects viremia, the opposite, MAYV infection followed by RRV challenge completely prevents viremia.
We have added the following to Section 3.2 to provide a potential explanation for this phenomena: “MAYV infection completely protected against delectable RRVTT viremia, whereas RRVTT infection only partially cross-protected against MAYV viremia (Fig. 2c). Although this might be interpreted as some form of one-way cross-protection [83], it should be noted that the “immunizing” viremias were (at their peak) ≈100 fold higher for MAYV than for RRVTT (Supplementary Fig. 4)”.
Point 2: Figure 5: Does JE/GETV vaccine provides complete protection against GETV virus? Some of the vaccinated mice (c) and (g) do not develop higher neutralization titers.
We have indeed assumed in the paper that the commercial GETV vaccine should provide complete protection against GETV; unfortunately, we are currently not in a position formally to confirm this in mice as we do not as yet have a mouse GETV model. GETV is a virus primarily of ungulates; we are seeking to develop a GETV mouse model and a number of GETV isolates are being tested (Rawle et al in preparation). Vaccine testing is only really appropriate if a GETV strain can be found that shows a reasonably rigorous viremia. Clearly, a GETV strain that shows a very poor viremia would be unrealistically easy to protect against.
We have added “(As we do not as yet have a mouse model of GETV, we were unable to verify that the commercial JEV/GETV vaccine can protect against GETV in mice)” to section 3.5.
Point 3: Is it possible to use create chimeric/mosaic proteins to generate broad protection as observed with viruses like influenza?
An interesting concept, although one might argue (extrapolating from Fig. 3c) that the trimeric heterodimer E1/E2 spikes are key to inducing effective neutralizing responses, and creating authentically folded spikes with chimeric/mosaic proteins might present a considerable hurdle. We do not feel it is appropriate to speculate about this issue in the manuscript as this would appear beyond the scope of the data presented.
Reviewer 2 Report
In this manuscript the authors conduct a systematic comparison of cross-neutralization and protection of arthritogenic alphaviruses. Ultimately use of a vaccine against one virus to protect against another would be a game-changer, increasing the potential impact of this study.
The breadth of experiments is impressive! The authors include IgG titers, neutralization titers, and viremia titers to lay out their cross-protective findings. In addition to their investigations, they characterized a non-adapted RRV strain (animal studies, sequencing, phylogenetics) in order to improve the significance of the studies. The data are quite clearly presented in the panels, which makes the results easy and straight-forward to follow. The interpretation of the results is appropriate and support the conclusions presented.
There are only a few minor comments regarding a couple of aspects of the manuscript.
- Section 2.10: Please add more detail for methods: PEG size, ultracentrifugation conditions (speed, time, temperature).
- Line 274: Suggest to add “Characterization of” RRVTT….infection in WT mice” to title of section
- All Figures: Suggest to enlarge all panels and move Legends to a new page for most of the figures.
- Suppl Fig 1b- Combining tables 1b and 1c would be a clearer way to present the information. Also, there are no amino acid numbers for the proteins.
- Figure 2: Overall nice correlation between identity % and cross-protection. RRV challenge often led to some breakthrough infection with low-level viremia and failed at complete protection. If the investigators had used the mouse-adapted RRV, would break through infections have been even more severe or produced higher viral titer?
- Overall, is low -level breakthrough viremia more or less significant in a WT model that leads to mild disease but mice recover? In the same vein as the IRF-/- deficient models, IFNAR mouse models exist for some of these diseases, and these mice have been shown to mount humoral responses. Therefore, it would be interesting to determine if the cross-protection remains complete in more sensitive models. Why did the authors choose a more severe model for RRVTT?
Author Response
Responses to Reviewer 2 Comments
Point 1: Section 2.10: Please add more detail for methods: PEG size, ultracentrifugation conditions (speed, time, temperature).
We have updated the methods sections with the PEG size and ultracentrifugation conditions.
Point 2: Line 274: Suggest to add “Characterization of” RRVTT….infection in WT mice” to title of section
We agree with this recommendation and have amended the title.
Point 3: All Figures: Suggest to enlarge all panels and move Legends to a new page for most of the figures.
We have complied with figure enlargements, but need to follow journal formatting guidelines with respect to the legends.
Point 4: Suppl Fig 1b- Combining tables 1b and 1c would be a clearer way to present the information. Also, there are no amino acid numbers for the proteins.
We have provided the amino acid numbers as suggested. We have, however, not combined the tables as this results in a very large table with many blank spaces (e.g. E3 and 6K have non-conservative changes exclusively) which means this figure would need 2 pages.
Point 5: Figure 2: Overall nice correlation between identity % and cross-protection. RRV challenge often led to some breakthrough infection with low-level viremia and failed at complete protection. If the investigators had used the mouse-adapted RRV, would break through infections have been even more severe or produced higher viral titer?
One might speculate that since mouse-adapted RRV replicates to higher viremias in mice (Figure 1) that breakthrough infections might be more common and/or reach higher viremias. Similarly, higher immunising viremias with T48 might also provide better cross-protection. This is an inherent variable that is difficult to control; however, such variability in viremias is also likely true in humans, so not inherently unrealistic and, we would argue, does not compromise the central finding that cross-protection is not reliably assured.
A key criticism, often repeated, is that the use of mouse-adapted viruses provides unrealistic answers. We have thus chosen to focus the studies on human isolates with no history of mouse adaptation.
Point 6: Overall, is low -level breakthrough viremia more or less significant in a WT model that leads to mild disease but mice recover? In the same vein as the IRF-/- deficient models, IFNAR mouse models exist for some of these diseases, and these mice have been shown to mount humoral responses. Therefore, it would be interesting to determine if the cross-protection remains complete in more sensitive models.
Cross-protection mediated by infection cannot be studied in these mice as infection results in mortality within a few days. We also provide evidence that cross-protection against mortality is not a good indicator or surrogate marker for cross-protection against viremia or arthritic disease (Fig. 6). Such studies would thus arguably provide unreliable answers to the key question of whether vaccines might provide cross-protection in humans.
Why did the authors choose a more severe model for RRVTT?
See Point 2 Reviewer 3.
Reviewer 3 Report
Here the authors aim to determine the level of cross-protection provided from various alphavirus vaccines. The authors have characterized a human isolate of RRV in wild-type C57Bl/6 mice which will serve as an important tool for the community. They then characterized cross-protection following infection of mice with CHIKV, MAYV, ONNV, and RRV. Most of the manuscript focuses on immunizing mice with a SCV-ZIKA/CHIK vaccine and testing for cross-protection against MAYV, ONNV, and RRV. The least amount of cross-protection was observed with RRV, which corresponded to the least conservation in E1/E2 sequences and more specifically in the E1/E2 receptor binding amino acids.
Major comments:
Figure 5 doesn’t fit well in this manuscript. It feels like a peripheral study. GETV wasn’t explored in the early figures. The authors don’t know the exact composition of the vaccine being tested. It is suggested that this figure be removed from the manuscript.
Figure 6: Given the results observed in Figures 1-4, what is the rationale for additional testing of the SCV-ZIKA/CHIK vaccine for RRV cross-protection but in an immune deficient model?
Minor comments:
Line 179: Please provide a reference for this statement “Viral preparations with even low levels of endotoxin contamination replicate poorly in wild- type mice”
The authors argue that there is limited commercial value in a vaccine that would offer cross-protection against multiple arthritogenic alphaviruses. However, it is important to note that mosquito-transmitted viruses have the potential to emerge in new regions of the world causing large outbreaks and even new diseases as has recently been observed with Zika virus. Mayaro virus is one pathogen that has caught the interest of the scientific community (PMID: 30254258).
Author Response
Responses to Reviewer 3 Comments
Point 1: Figure 5 doesn’t fit well in this manuscript. It feels like a peripheral study. GETV wasn’t explored in the early figures. The authors don’t know the exact composition of the vaccine being tested. It is suggested that this figure be removed from the manuscript.
We included Figure 5 as GETV is the closest phylogenetically to RRV and, importantly, is the only alphavirus with a commercially available vaccine. We have added “…and currently represents the only arthritogenic alphavirus for which a vaccine is commercially available.” to section 3.5 to emphasise this issue.
To omit this vaccine would seem to us to be quite inappropriate as the key finding of this study revolves around the commercial viability of reliance on cross-protection, with a similarly fixed RRV vaccine having also been tested in a phase III clinical trial. We have a large body of work (Rawle et al, in preparation) that investigates GETV more fully, but this GETV focused paper covers a range of primarily veterinary issues, inappropriate for the current manuscript.
We also wanted to compare and contrast between a live recombinant vaccine (SCV-ZIKA/CHIK) and the formalin-inactivated vaccine (JEV/GETV) and their significantly contrasting abilities to induce cross-reactive neutralizing antibodies (Supplementary Fig 8c). The composition of the vaccine is described by the company; the exact composition (like nearly all commercially available vaccines) is commercial in confidence. This should surely not preclude their use in research endeavours.
Point 2: Figure 6: Given the results observed in Figures 1-4, what is the rationale for additional testing of the SCV-ZIKA/CHIK vaccine for RRV cross-protection but in an immune deficient model?
A large number of researchers use such models to test interventions, with protection against mortality often used as an indicator of efficacy [45,52,89-91]. We have added the underlined phrase to highlight the rationale for 3.6. We have also added phrases to the end of this section to highlight that this vaccination failed to protect against viremia in wild-type mice (Fig. 4j) and to clarify the take home. Our data would argue that such commonly used models might show protection against mortality, without protecting against viremia or arthritic disease. Mortality read outs in these models are thus poor surrogate markers for cross-protection (and by implication protection) against viremia or arthritic disease. By implication, protection against alphavirus mortality might be easier to achieve in such mice and may thus, for instance, provide unrealistic optimism about the utility of a particular vaccine for cross-protection.
Point 3: Line 179: Please provide a reference for this statement “Viral preparations with even low levels of endotoxin contamination replicate poorly in wild- type mice”
This is our experience with alphaviruses, so we have added “In our experience, alphaviral….”. We have not published this as it is relatively well known that endotoxins induce IFNa/b and that alphaviruses are very sensitive to IFNa/b.
Point 4: The authors argue that there is limited commercial value in a vaccine that would offer cross-protection against multiple arthritogenic alphaviruses. However, it is important to note that mosquito-transmitted viruses have the potential to emerge in new regions of the world causing large outbreaks and even new diseases as has recently been observed with Zika virus. Mayaro virus is one pathogen that has caught the interest of the scientific community (PMID: 30254258).
We intended this concept to be implied by the word “current” followed by 2 references talking about two potentially emerging viruses (RRV which caused an outbreak in the Pacific Islands and Zika virus quoting PMID: 30254258). We have added a phrase to make this clearer.
Reviewer 4 Report
The manuscript, “Arthritogenic alphavirus vaccines: serogrouping versus cross-protection in mouse models” offers a comprehensive evaluation of the ability of wild-type arthritogenic alphavirus infection and two different alphavirus vaccines to cross-protect against heterologous alphavirus challenge in several mouse models. The manuscript is generally well-written and the science is sound, with studies using a sufficient number of mice for the statistics employed. However, the manuscript could potentially be improved by addressing the following:
- Several minor typographical errors were observed (overlooked punctuation, misspellings, etc) throughout, so the manuscript could benefit from additional proof-reading.
- Line 13, it is unclear what the authors mean by “mice were infected…s.c. into the top/side of each hind foot as described previously.” The general approach to infecting via this route is to inject what this reviewer considers to be the “bottom” (or ventral) side of one of the rear footpads. This can be clarified by using anatomic designators.
Thank you for the opportunity to review this manuscript.
Author Response
Responses to Reviewer 4 Comments
Point 1: Several minor typographical errors were observed (overlooked punctuation, misspellings, etc) throughout, so the manuscript could benefit from additional proof-reading.
We have undertaken additional proof-readings of the manuscript.
Point 2 Line 13, it is unclear what the authors mean by “mice were infected…s.c. into the top/side of each hind foot as described previously.” The general approach to infecting via this route is to inject what this reviewer considers to be the “bottom” (or ventral) side of one of the rear footpads. This can be clarified by using anatomic designators.
This is a comment we have frequently received and so we have added a new supplementary figure with photographs of the foot injection to provide complete clarity (new Supplementary Fig. 1). Much of the confusion comes from the definition of “footpad”. Our AEC considers the footpads the keratinized pads that contact the ground when the animal is walking. Injection into the footpad or any walking surface is not allowed by our AEC, as this can cause discomfort and gait abnormalities. Others consider the bottom of the foot the footpad. Yet others consider the whole palm of the foot as the footpad.